# The Effect of an Innovative Biofeedback SKOL-AS^®^ Treatment on the Body Posture and Trunk Rotation in Children with Idiopathic Scoliosis—Preliminary Study

**DOI:** 10.3390/medicina55060254

**Published:** 2019-06-07

**Authors:** Anna M. Kamelska-Sadowska, Halina Protasiewicz-Fałdowska, Lidia Zakrzewska, Katarzyna Zaborowska-Sapeta, Jacek J. Nowakowski, Ireneusz M. Kowalski

**Affiliations:** 1Clinic of Rehabilitation, Provincial Specialist Children’s Hospital in Olsztyn, 18A Żołnierska Street, 10-561 Olsztyn, Poland; i.kowalski@wp.pl; 2Department of Rehabilitation, Faculty of Medicine, Collegium Medicum, University of Warmia and Mazury in Olsztyn, 18A Żołnierska Street, 10-561 Olsztyn, Poland; prohalina@wp.pl (H.P.-F.); dr.zaborowska@gmail.com (K.Z.-S.); 3HUMANUS Centre of Rehabilitation, 15B Kanta, 10-691 Olsztyn, Poland; lidia.zakrzewska@onet.pl; 4Department of Ecology and Environmental Protection, University of Warmia and Mazury in Olsztyn, 3 Lodzki Square, 10-727 Olsztyn, Poland; jacek.nowakowski@uwm.edu.pl

**Keywords:** trunk rotation, idiopathic scoliosis, scoliotic posture, children, SKOL-AS^®^, SpinalMeter^®^, biofeedback

## Abstract

*Background and Objectives:* The deformity in idiopathic scoliosis (IS) is three dimensional and effective correction involves all three planes. Recently, the biofeedback method has been implemented in the treatment of scoliosis. The aim of this study was to evaluate the effectiveness of an innovative biofeedback SKOL-AS^®^ postural training among children with scoliosis. *Materials and Methods:* The target population for this study was 28 patients (25 girls and 3 boys) aged between 5 and 16 years old diagnosed and treated with progressing low-grade scoliosis. The postural diagnosis consisted of anthropometric measurements, posterior–anterior X-ray imaging, SpinalMeter^®^ postural assessment and the angle of trunk rotation (ATR) assessment. The SKOL-AS^®^ treatment comprised of 24 sessions conducted in lying and sitting positions, two times a week. *Results:* It has been shown that the postural training resulted in the decrease in the ATR value (pre- vs. post-exercise in younger: 5.55 vs. 3.0 and older patients: 5.2 vs. 3.0). The increase in height of the subjects seemed to confirm a positive effect of SKOL-AS^®^ elongation treatment. In the posterior view, a statistically significant decrease in shoulder asymmetry in the sitting position in younger children has been observed. In the anterior view, the changes in the head position (based on mouth and eye symmetry) have been observed. The statistically significant increase in acromion–heel, acromion–iliac crest and posterior superior iliac spine (PSIS)–heel length values has been shown in younger children on the left side of the body. After treatment, older subjects had higher acromion–iliac crest and PSIS–heel values on the left side of the body. On the right side only PSIS–heel length was higher. In a sitting position, only a small increase in acromion–iliac crest length value has been observed. *Conclusions:* The SKOL-AS^®^ biofeedback method could teach good postural habits and teach patients the auto-correction of the spine.

## 1. Introduction

Scoliosis is a three-dimensional (3D) spinal deformity consisting of a lateral curvature in a frontal plane (with a Cobb angle of 10° or more), sagittal deformity and rotation of the vertebrae in a transverse plane [1,2].

The Society of Scoliosis Orthopedic Rehabilitation and Treatment (SOSORT) guidelines provide clear, scientific indications as to what type of treatment is appropriate for patients with scoliosis [3]. The therapy for adolescent idiopathic scoliosis (AIS) depends on the patients’ age, the degree of curvature, direction and pattern of the spinal curve, type of scoliosis, maturity status and also the risk of progression [3].

The most commonly used treatments include observation, exercise [4], orthotic management (bracing) [5], as well as surgical correction with or without fusion [6].

Nowadays, the feedback method has gained much importance in the field of medicine and rehabilitation. The latest scientific achievements [7] have resulted in a new trend emerging in rehabilitation. 

Biofeedback is a nonmedical process that involves the measuring of specific and quantifiable bodily functions of a subject, such as brain wave activity, blood pressure, heart rate, skin temperature, sweat gland activity and muscle tension, thus conveying the information to the patient in real-time. The basic aim of biofeedback therapy is to support a patient in realizing his/her self-ability to control specific psychophysiological processes [8]. 

The biofeedback is based on the nervous system stimuli, which determines its high effectiveness. 

Recently, the biofeedback method has been implemented in the treatment of scoliosis [9]. A modified pressure biofeedback unit has been used. Participants performed segmental spinal movements that primarily involved segmental spinal stabilizing muscles with graded and sustained muscle contraction against/off a pressure cuff from baseline to target pressures and then maintained for 1 min [9]. In another study, the specific sensory-signal system has been successfully used as an advanced biofeedback therapy method that enhanced self-correction of the undesirable posture. This has also been used as a diagnostic tool. The biomechanical analysis of the participants has been based on the Spinalmouse^®^ device. After a 4 h session with sensory-signal system, a significant (*p* < 0.05) improvement of their posture has been shown [10]. 

In addition, the biofeedback has been applied in a number of research studies using surface electromyography (sEMG) as an instrument for muscle rehabilitation. It has been concluded that with regular practice of the corrected positions, those with AIS can use motor learning to achieve a more balanced posture. Consequently, the findings can be used in less intrusive early orthotic intervention and provision of care to those with AIS [11].

The SKOL-AS^®^ method is an innovative treatment based on the corrective work and postural training learned by patients after receiving a visual signal from specific manometers. The idea of the SKOL-AS^®^ therapy and the device was introduced as treatment in 2011, when the Patent Office approved the device for physiotherapy of spine disorders (Urządzenie do monitorowania ćwiczeń rehabilitacyjnych wykonywanych przez pacjentów do rehabilitacji schorzeń kręgosłupa; Patent number: PL 221 322 B1). Furthermore, an additional patent was granted in Munich by the German Patent Office (number 10145999) on 15 April 2003. The effectiveness of biofeedback on the position taken during habitual standing as well as sitting is gaining more and more interest. Thus, in this study the therapy consisted of exercises during lying and in a sitting position, as well as corrective tasks in front of the mirror in the standing position. 

The aim of the study was to evaluate the effectiveness of an innovative biofeedback SKOL-AS^®^ postural training among children with scoliosis. 

We have hypothesized that control of the scoliotic curves progression would come from the continuous training of spinal muscles through biofeedback. The active forces through the muscular contraction could be accomplished and the postural training could be performed. 

## 2. Materials and Methods

### 2.1. Participants

The target population for this study was 28 patients (25 girls and 3 boys) aged between 5 and 16 years old, diagnosed and treated with progressing low-grade scoliosis at the Humanus Centre of Rehabilitation in Olsztyn, Poland and at the Stanislaw Popowski Regional Specialized Children’s Hospital in Olsztyn, Poland (Table 1).

The characteristics of the population studied are shown in Table 1. There were statistically significant differences (*p* < 0.05) between younger and older participants, but no differences between gender; therefore, the studied population was divided into two groups: (A) juvenile scoliosis (between 6 and 11 years old; *n* = 11; 41% of all participants) and (B) adolescent scoliosis (from 12 to 16 years old; *n* = 16; 59% of all participants). 

In this study, 64% of the children had lower Risser stage (RS), which is perceived to be associated with higher progression incidence (in particular: RS 0–2 = 64% of all participants; RS 3–4 = 36% of all participants). 

The inclusion criteria consisted of: first grade scoliosis confirmed in the clinical and radiological assessment (according to Cobb classification) with Cobb’s angle between 10–20°. Patients had no history of brace treatment, no co-morbidities affecting the course of scoliotic deformation such as genetic defects, neuromuscular disorders, metabolic disorders and history of severe trauma. Patients who had been treated previously; who did not comply with exercise recommendations or prematurely stopped the exercise; who were simultaneously using another method of correction; and with RS more than 4 were excluded from the study. The qualification of patients’ posture as scoliosis were based on the guidelines of the Society on Scoliosis Orthopaedic and Rehabilitation Treatment (SOSORT) [3]. The patients had thoracic (*n* = 4), thoracolumbar (*n* = 15), lumbar (*n* = 3) and double curve (*n* = 6) deformation.

### 2.2. Experimental Design

All the procedures performed in the study involving human participants conformed to the ethical guidelines of the 1975 Declaration of Helsinki (and its later amendments or comparable ethical standards, revised in 2013) and followed the Adapted Physical Activity (APA) Ethics Standard [12]. The protocol was approved by the Ethics Committee of Stanislaw Popowski Regional Specialized Children’s Hospital in Olsztyn, Poland (number of approval: ZE/1/2018/WSSD; date of approval: 10 October 2018). The experiment was conducted with the understanding of each subject. All subjects as well as their parents gave written informed consent before children’s participation in this study.

### 2.3. Methods

In this study, both radiographic and anthropometric measurements have been chosen for the clinical assessment to evaluate the efficacy of the postural training. Basic postural diagnosis consisted of anthropometric measurements, a posterior–anterior (P–A) X-ray imaging (only in preliminary examination), SpinalMeter^®^ postural assessment and the angle of trunk rotation (ATR) assessment. The treatment has been based on an innovative biofeedback SKOL-AS^®^ method.

#### 2.3.1. Anthropometrics

Children’s body weight (after removal of shoes and heavy clothing) was measured to the nearest 0.1 kg. Height was measured to the nearest 0.01 m using the standard portable column scale (Seca 217, Spoland Wagi Elektroniczne, Warsaw, Poland). Body mass index (BMI; kg m^−2^) was calculated (to one decimal place) as:
$$BMI=\frac{weight}{height^{2}}=[\frac{kg}{m^{2}}]$$

Spine length has been measured to the nearest 0.1 cm from C7 to S1 spinous processes (palpated by the physician) using standard measuring tape in the sitting position.

#### 2.3.2. X-ray Imaging

A posterior-anterior X-ray (P-A), in standing position, of the general spine with vision of the iliac crests and femoral heads was performed once before the SKOL-AS^®^ treatment. P-A Cobb’s angles were measured using Cobb’s method from standing P-A radiographs [13].

The Risser stage was assessed according to the European Risser Staging System [14]. An X-ray of the patients was taken a day before the SpinalMeter^®^ examination. 

#### 2.3.3. SpinalMeter^®^ Calibration and Posture Assessment

SpinalMeter^®^ was invented by Bellavigna, Gianluca in Terni (Italy) and the European Patent was awarded by the European Patent Office (EP 3225155 A1; application number: 08425006.7) [15]. The validation of SpinalMeter^®^ biometrical assessment has been proven and the most reliable results were obtained for length measurements of acromion (ac)-popliteal fossa (PF) and ac-posterior superior iliac spine (PSIS) (for ac-PF: coefficient of variation (CV) = 0.29; variance (V) = 9.81; for ac-PSIS: CV = 0.45; V = 3.47). The lowest coefficient of variation (CV = ~0.30; V = ~0.50) had scapular asymmetry both in standing and sitting positions as well as pelvic asymmetry in sitting position [16]. The calibration of SpinalMeter^®^ was performed on the day of research, before patients’ measurements. It was based on the precise determination of four points on the calibration platform. The real-time image was done and the middle line was set.

Anthropometric points on the body has been defined by landmarks (Maestrale^®^ Italy markers) that point out the spine’s position and limb’s length. The latter and other readings (e.g., surface of the triangles of the size)were used for a complete evaluation of the subjects in different positions (Figure 1). The precise fit of the landmarks was done by specific HSL (hue, saturation, lightness) linear correction when the displacement (shift) of the skin has been considered (Figure 2). The anatomic indexing points are shown in Figure 1. Moreover, for better precision, specific points were additionally drawn on the patient’s body with a marker pen. In this study, only selected positions of the body, as well as asymmetry and length measurements, were analyzed.

The postural biometrical assessment consisted of the palpation and the determination of anatomic points by the same evaluator [Physical Medicine and Rehabilitation (PM&R) physician and physiotherapist] at:
posterior view in standing:
(1)C7—cervical spine(2)Th4—thoracic spine(3)Acromion (right and left side)(4)Angulus inferior scapulae (right and left side)(5)L1—lumbar spine(6)Olecranon—elbow (right and left side)(7)Iliac crest (right and left side)(8)PSIS—posterior superior iliac spine (right and left side)(9)Popliteal fossa—the hollow at the back of the knee (right and left side)(10)Heel (right and left side)posterior view in sitting position: all above mentioned points without 9 and 10anterior view:
(1)Right and left eye(2)Right and left side of the mouth(3)Clavicle (right and left side)(4)ASIS—anterior superior iliac spine (right and left side)(5)Radial styloid process (right and left side)(6)The middle of the knee (right and left side)(7)Medial malleolus (right and left side)(8)First and fifth metatarsal bones (right and left side)

Once the reflective markers were in place the patient was positioned for the digital photo examination, in the frontal plane with the arms hanging at the side of the body and the feet and knees together, according to the natural stance of the patient. The exact location of the feet has been ensured by the specific “calibration” platform.

The diagnostic procedure consisted of:palpation and the use of reflective markers to mark the spinous processes and specific anatomical points;acquisition of digital images;processing of the image and the tag applied through a specialist software (SpinalMeter^®^ Evolution v6.14);numerical visualization of the patient’s spine;visualization with bidimensional image of the spine’s curvature;measurement of upper and lower limb.

The SpinalMeter^®^ system consists of: a personal computer system (Acer Aspire E17; CPU Intel (R) Core (TM) i5-5200U 2.2 GHz, 64 bit and 8 GB RAM; operating system Windows 10 Education; monitor with 1024 × 768 Pixel resolution; Maestrale Information Technology Srl, Terni, Italy); a stand with the camera (Canon EOS 1200D with 18 megapixel CMOS; Maestrale Information Technology Srl), a “calibration” platform (42 × 42 cm), and a stand-platform connector (the distance between the patients heel and the camera = 243 cm). The whole portable system used in this study is shown in Figure 3. In this study, the standing and sitting positions have been considered. The following ten measurements in a short period of time (6 s) were taken by two professional experts.

#### 2.3.4. The Angle of Trunk Rotation (ATR) Assessment

The measurements of the ATR were performed using a scoliometer during Adams forward bending test by the two evaluators/diagnosticians [17]. It is known that the measured rib hump is directly related to spinal rotation and rib deviations. The reliability of the measurements obtained with the scoliometer was determined as very good to excellent in a previous study [18]. Thus, in this study the scoliometer was used to analyze the axial rotation of the trunk (i.e., ATR) in the studied group of patients. The scoliometer was placed over spinous processes of the back and was drawn along them to measure the axial trunk rotation. During the first measurement the spinous process with the highest value of ATR was marked with a waterproof marker. This space was used during the next measurements in order to reproduce the same level as in the preliminary examination. The ATR evaluation performed using the scoliometer with the participants standing in trunk flexion is shown in Figure 4.

#### 2.3.5. Control Examination

The control examination (after 3 months SKOL-AS^®^ therapy) was based on clinical assessment only (anthropometrics, SpinalMeter^®^ examination and ATR assessment). This was due to the fact that multiple X-ray exposures for the experiment would be too burdensome for the young patients. This would not be acceptable for ethical reasons.

#### 2.3.6. SKOL-AS^®^ Device Therapy

The treatment consisted of a short warm-up of the thoraco-lumbar back area, using simple exercises as cat-cow, plank, and exercises with TheraBand^®^ resistance bands. In the very beginning, the stabilizing forces at the pelvis region were applied. Then the external corrective forces on the scoliosis curves in the frontal plane were applied. These were generated by a designed, experimented and innovative system (SKOL-AS^®^) (Figure 5 and Figure 6). The specific de-rotation forces were additionally applied during training in the lying position. The lying position and de-rotation cushions are shown in Figure 6. The patient was asked to push into the cushions on the concave side of the curve (passive lateral bending) (e.g., 4 s, 5–10 times). Simultaneously, the pressure to the apex of the scoliosis curve and the de-rotation force were applied. Next, the patient was asked to push on the other side of the scoliotic curve and the pressure was measured on the specific manometers (Figure 5). The external force was applied individually for each patient. The magnitude of this force was performed by the physiotherapist who worked with the patient. It was ca. 40% of maximal force, which could be performed by an individual patient. During exercise in the sitting position, specific elongating cushion was used for spine elongation. The effective exercise work performed by the patient included the deep muscles exercise with visual biofeedback starting from 40 mmHg to 60, 70, 80, or 100 mmHg on the manometer scale. This allowed the patient to control his/her posture. After a 30 min session the patient was asked to perform some core stability exercises to stabilize the effect of the SKOL-AS^®^ therapy.

The SKOL-AS^®^ treatment consisted of 24 sessions (three months of therapy, two times a week), each lasting 30 min:

Lying position:
1st–2nd session—15 min biofeedback training + 15 min SKOL-AS^®^3rd–4th session—10 min biofeedback training + 20 min SKOL-AS^®^5th–8th session—30 min SKOL-AS^®^

Sitting position:9th–24th session—30 min SKOL-AS^®^

During the last four sessions the specific posture auto-correction in the standing position was performed.

#### 2.3.7. The Learning of Posture Correction during Standing and Sitting Positions

The requirements of the standing and sitting positions were used according to the recommendations shown in McKenzie (2011) [19]. The patients were guided to perform the posture corrected by the physiotherapist as follows:uggested sitting position: the head and ankles should be straight, shoulders and hips are level, kneecaps face the front, and the chin should be parallel to the floor and aligned with the ears. The lower back should be slightly bent forward to support the body with no extra weight distributed onto the spine.Suggested standing position: the head and ankles should be straight, shoulders and hip are level, kneecaps face the front, the head and knees are straight, and the chin should be parallel to the floor and aligned with the ears. The lower back should be slightly bent forward with the aid of the chest, stomach, and buttock muscles.

#### 2.3.8. The Statistical Methods

Statistical analysis was performed using Statistica 13.0 software (TIBCO Software Inc. 2017, Krakow, Poland) [20]. All measurements are shown as mean ± standard deviation (SD). The measurements are given with the accuracy that they were performed. In the case of estimators (e.g., average and other moments of distribution), the accuracy is shown by one decimal place more and in the case of standard error of mean (SEM) or SD, two decimal places more than original measurements, because the value of estimators (descriptive statistics from sample) shows the accuracy of the average estimation. The statistical significance has been set at *p* < 0.05. There were no gender differences in the characteristics of the population studied (*p* > 0.05). Thus, boys and girls were combined into one group. Due to the diversity of characteristics in the age groups, the analysis was performed taking two age classes, 5–11 and 12–16 years old, into account. The asymmetry of the selected features was measured considering the direction of asymmetry—left-sided or right-sided—and changes in the degree of asymmetry for measurements before and after SKOL-AS^®^ postural training. They were calculated considering the direction of asymmetry. Distributions of differences in the values of repeated measurements and distributions of analyzed features in the studied groups were compared with the normal distribution by the Shapiro–Wilk test. Depending on the assumptions of test functions, the factorial mixed model of variance analysis (factorial mixed ANOVA) with repeated measurements (before and after SKOL-AS^®^ postural training) and the classification variable (age classes) were used for analysis. In the case of significant interactions, the results of comparisons between individual groups were based on the Tukey test, while in other cases the means were compared separately in the age groups with a one-dimensional parametric test (T-paired test) or non-parametric test (Wilcoxon test) depending on analysis conditions.

## 3. Results

### 3.1. The Characteristics of the Population Studied 

The characteristics of the population before and after treatment (mean and SD) are shown in Table 2. The patients learned easily how to make the necessary postural adjustments during SKOL-AS^®^ postural training. Even one child (aged 5 years old) precisely followed the exercise tasks. After 24 sessions all patients obtained knowledge necessary for auto-corrections of scoliotic curve and elongation of the spine during their daily life. They gained precise knowledge about their deformation and dysfunctions. The statistically significant increase in height of patients was observed. However, there were no statistically significant differences in spine length before and after SKOL-AS^®^ postural training (Table 2). 

### 3.2. The Effect of SKOL-AS^®^ Postural Training on the Angle of Trunk Rotation

The angle of trunk rotation (ATR) values before and after therapy are shown in Table 3. The statistically significant differences before and after exercises using the SKOL-AS^®^ device were shown in younger as well as in older children. Postural training combined into 24 sessions in lying and sitting positions had a positive impact on trunk rotation and resulted in a decrease in the ATR value. The effect of exercise using the SKOL-AS^®^ device during three months of therapy on the anthropometric points asymmetry in standing and sitting positions is shown in Table 4 and Table 5. If the symmetry was shifted to the right side of the body the values were shown as positive (+) and if the symmetry was shifted to the left side then negative (−) values were observed.

In the posterior view, the statistically significant decrease in shoulder asymmetry in the sitting position in younger children has been shown. The decrease in shoulder asymmetry has also been shown in children aged 12–16 years old. However, these changes were not statistically significant. The decrease of scapular asymmetry has been shown in older children in the standing as well as sitting position. However, in younger subjects the scapular asymmetry after SKOL-AS^®®^ training increased significantly. This could be associated with the changes which occurred in other regions of the body (e.g., pelvis or PSIS) whose symmetry changed direction after the exercise. Moreover, this increase could be induced by lower age and lower understanding of the auto-corrections in this age group. The directional change of pelvic symmetry was observed only in older children in the sitting position. The changes observed in PSIS symmetry was not statistically significant. 

In the anterior view, the changes in the head position (mouth and eye symmetry) were observed. However, these changes were not statistically significant. Only the increase in ASIS asymmetry was significant (higher after SKOL-AS^®^ treatment). Moreover, much higher asymmetry of ASIS was observed in the younger subjects. 

The changes in length values on the right and left side of the body before and after training is shown in Table 6. The statistically significant increase in acromion–heel, acromion–iliac crest and PSIS–heel length values is shown in younger children on the left side of the body. On the right side only PSIS–heel length was higher after exercise. Older subjects after treatment had higher acromion–iliac crest and PSIS–heel values on the left side of the body. On the right side only PSIS–heel length was higher. The angle of heel–popliteus fossa decreased, however the changes were significant only on the right side of the body. In the sitting position, only a small increase in acromion–iliac crest length value was observed (Table 7).

## 4. Discussion

The deformity in idiopathic scoliosis (IS) is three dimensional in nature and effective correction involves all three planes. The SKOL-AS^®^ is a relatively new biofeedback device and its effectiveness in the treatment of IS has not yet been shown. What should be highlighted is that this system works on postural muscles in a three-dimensional manner.

In this study, all patients gained specific knowledge, which improved their understanding of spine deformation and the auto-correction of their body. 

After three months of training, statistically significant changes in the height of patients was observed. This seems to confirm the positive effect of elongation exercises used in SKOL-AS^®^ device treatment. The decrease in trunk rotation was also observed (see Table 3). 

A statistically significant increase in the height of patients was observed. Previously, it was shown that the deformity progresses most rapidly during the period of fast growth [21] and growth in height is a sine qua non for the development of scoliosis [22]. Other authors have shown that idiopathic structural scoliosis often makes its first structural appearance within three periods of life - periods during which growth in height is rapid, viz. 0–3 years (infantile scoliosis), 5–8 years (juvenile scoliosis) and after 10 years (adolescent scoliosis). The last group is the largest [23]. In this study, the last two groups were included. Thus, the increase in height could be the result of elongation after treatment as well as the natural growth of the children. On the other hand, as a consequence of the scoliosis progression, the height of subjects decreased, which was observed in some cases.

Moreover, changes in the anthropometric points symmetry as well as length values were different after the therapy. This proves that positive changes were observed after exercises using SKOL-AS^®^ device. 

In this study, manometry-guided biofeedback was used. This helped the patients to understand the direction of spine correction. Other authors have used a specific device with tone alarm signaling when poor posture occurred. It has been shown that a long-lasting active spinal control could be achieved through the patient’s own spinal muscles [10]. 

The latest studies have shown that a specific tank top equipped with sensors can motivate patients to adopt a more active role, thus more effectively improving their control and coordination of movement and daily posture [7]. The effectiveness of this training was evaluated by using sEMG signals and 3D ultrasonic imaging of the spine. Thirty sessions of postural training proved to be enough to train a patient’s sitting posture. The patient’s body was relatively more balanced, and this involved a lower degree of muscle activity in terms of sEMG signals compared to their circumstances prior to the training. Moreover, improvement of the body posture by means of 3D ultrasonic imaging of the spine was observed. 

It was shown in this study that by means of a non-invasive postural training good postural habits can continue in daily life. The SKOL-AS^®^ device could cure the scoliosis curve and prevent further spinal deformity progression. However, not every variable was changed as it was expected. Based on the results, it was assumed that the SKOL-AS^®^ treatment could be applied only in some specific deformations or points of body and the exercise could be performed with children who precisely understand the force, pressure and the direction how to make the corrections. 

There were many positive values of this research. The use of the SKOL-AS^®®^ device helped patients with controlling their posture (e.g., palpating their spinous processes and the working muscles). Moreover, the precise work restricted in the actual place where the deformation occurs helped the patient to better understand the force and the direction which should be applied to correct the spine. The use of manometers and cushions gave the opportunity to have hands free to work with the patients. Sometimes the use of a Thera-Band as well as PNF (Proprioceptive Neuromuscular Facilitation) and the Lehnert–Shrot method could be performed along with the SKOL-AS^®^ correction device. 

On the other hand, there were also some limitations. The sample size (mainly because of financial reasons) was rather too small to make strong conclusions. Furthermore, this study lacked a control group, thus it was difficult to draw meaningful conclusions from the study and produce an erroneous results. On the other hand, this research was a pilot study with preliminary results, which was the intention of the authors. Therefore, future studies should include more participants. Moreover, two groups and a control group (SKOL-AS^®^ treatment vs. other methods) could be implemented in future research. The prospective controlled trial of pairs of patients with idiopathic scoliosis matched by sex, age, Cobb angle and curve pattern could be done in future research. The effect of different exercise programs in the treatment group [exercises designed for scoliosis (e.g., FITS – Functional Individual Therapy of Scoliosis) + SKOL-AS® exercises] and the control groups (exercises designed for scoliosis or FED - esp. Fijación, Elongación, Desrotación; eng. Fixation, Elongation, Derotation treatment) could be compared. Moreover, in this study the brace introduced during research program was a limiting factor. In a future study the comparison of brace vs. non-brace exercise programs could be compared. The follow-up after e.g., 3 months after therapy using X-ray imaging could also be considered.

## 5. Conclusions

The SKOL-AS^®^ biofeedback method uses three-dimensional exercises, thus it seems to be a good way to treat scoliotic spine.The decrease in trunk rotation and some anthropometric points asymmetry and the increase in length values proves the effectiveness of the SKOL-AS^®^ treatment. However, some results are questionable and need verification.The SKOL-AS^®^ biofeedback method could help patients learn good postural habits and auto-correction of the spine.

## Figures and Tables

**Figure 1 medicina-55-00254-f001:**
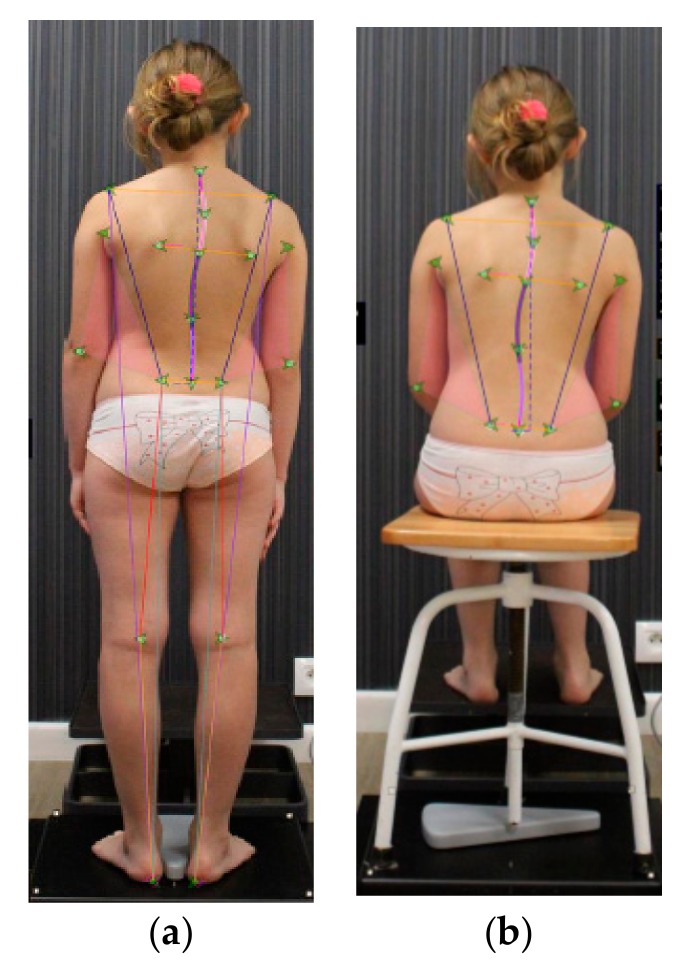
The anatomic indexing points and the position of marker points during postural biometrical assessment by SpinalMeter^®^ in (**a**) standing and (**b**) sitting positions (patient JM; age = 8 years old, own data, the photo was taken at the Humanus Centre of Rehabilitation in Olsztyn).

**Figure 2 medicina-55-00254-f002:**
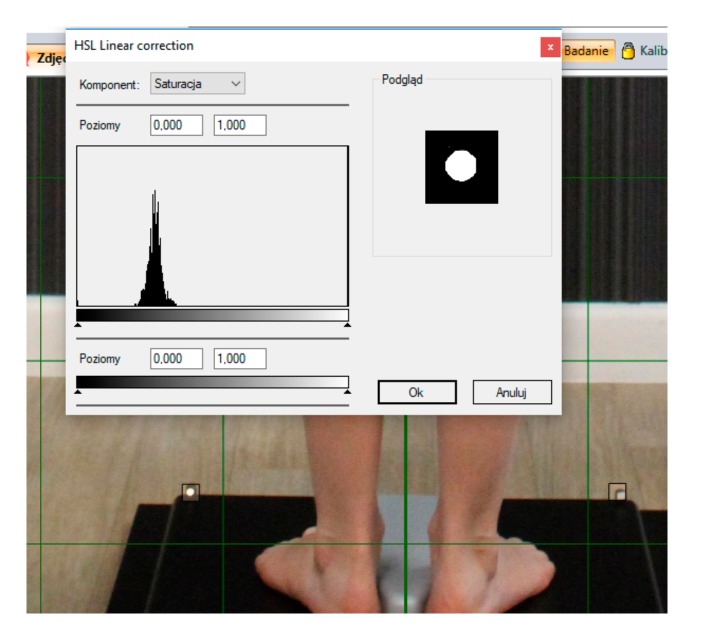
Linear correction after pointing the landmarks by a researcher (own data; the photo was taken at the Humanus Centre of Rehabilitation in Olsztyn).

**Figure 3 medicina-55-00254-f003:**
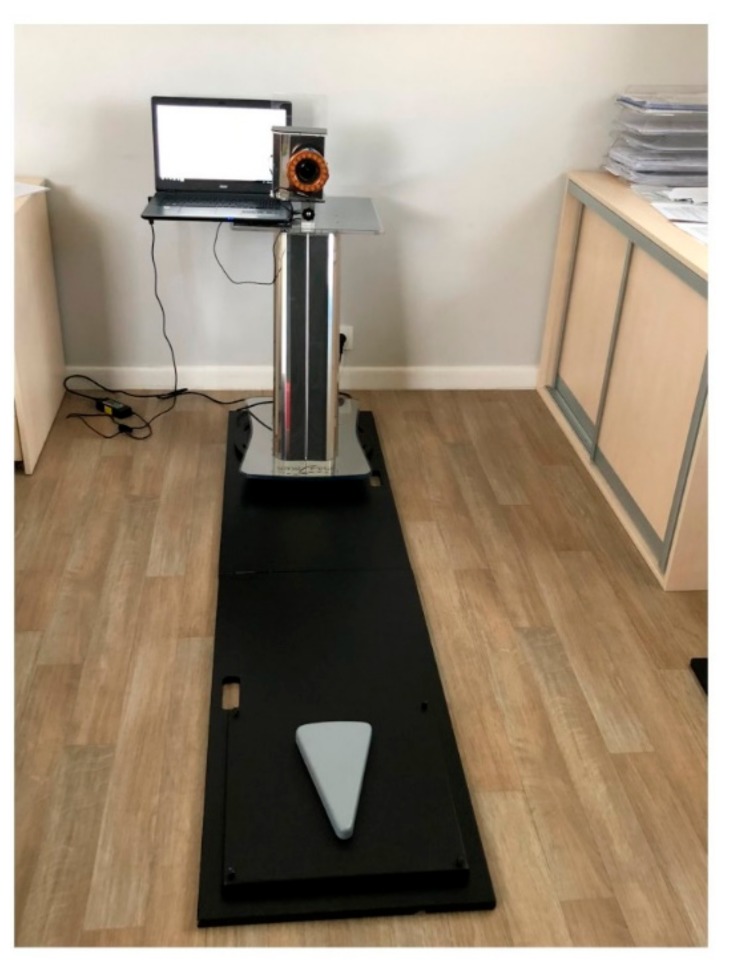
SpinalMeter^®^ portable equipment (own data, the photo was taken at the Humanus Centre of Rehabilitation in Olsztyn, the platform has been shortened for better vision).

**Figure 4 medicina-55-00254-f004:**
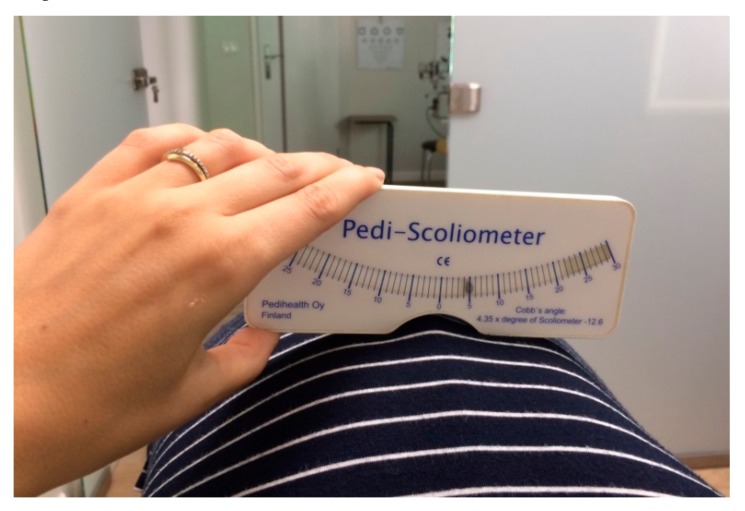
The evaluation of the angle of trunk rotation (ATR) during Adams forward bending test (own data, the photo was taken at the Humanus Centre of Rehabilitation in Olsztyn).

**Figure 5 medicina-55-00254-f005:**
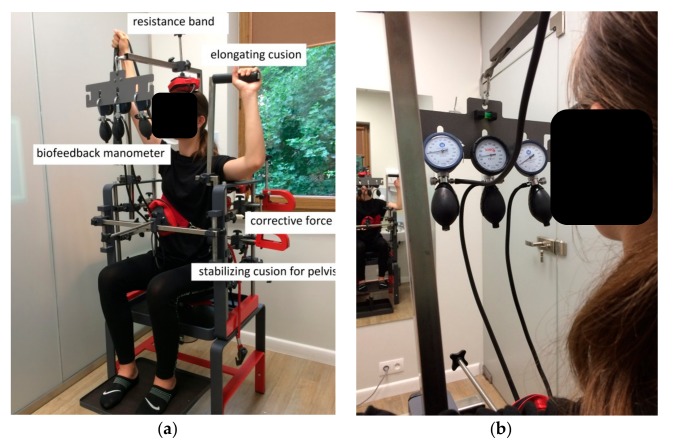
(**a**) The SKOL-AS^®^ device—patient’s postural training in a sitting position (patient SA, age = 15 years old, left thoracolumbar scoliosis; own data); (**b**) manometers used for biofeedback treatment (right side). Own data, the photo was taken at the Humanus Centre of Rehabilitation in Olsztyn.

**Figure 6 medicina-55-00254-f006:**
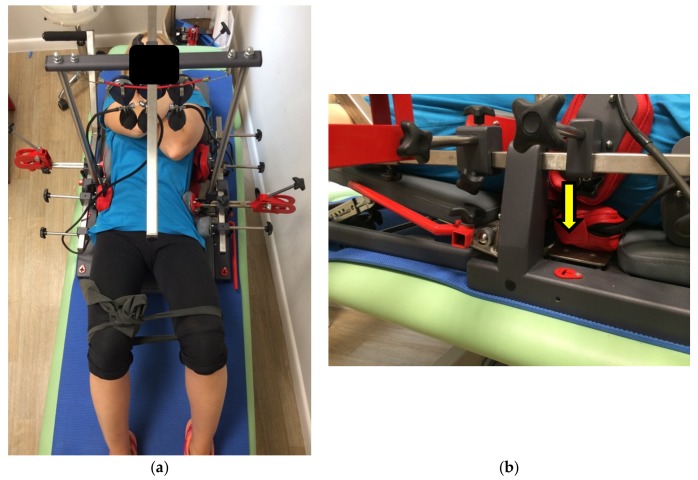
(**a**) The SKOL-AS^®^ device—patient’s postural training in a lying position (patient KK, age = 12 years old, left lumbar, right thoracic scoliosis; own data); (**b**) de-rotation cushions in lying position on the right side of the body have been magnified [the same position as in (**a**)]. The direction of the force applied by the patient has been shown by an arrow. Own data, the photo was taken at the Humanus Centre of Rehabilitation in Olsztyn,

**Table 1 medicina-55-00254-t001:** The characteristics of the population studied.

Variables/Age (Years)	Gender Man/Woman (Number)	Cobb (°)	ATR (°)	Risser Stage	Age (Years)	Height (m)	Body Mass (kg)	BMI (kg/m^2^)
5–11	2/9	14 (10–28)	1 (0–3)	0 (0–3)	8 (5–11)	1.41 (1.19–1.74)	34.0 (22.0–60.0)	16.0 (14.3–19.8)
12–16	1/15	13 (6–23)	1 (1–3)	2 (0–4)	13 (12–16)	1.61 (1.51–1.74)	54.0 (34.0–61.5)	18.9 (14.2–24.1)

Data shows: Me–Median (min.–max.).

**Table 2 medicina-55-00254-t002:** Mean and standard deviation (SD) of the characteristics of the population studied before and after SKOL-AS^®^ treatment.

Variables Age (Years)	Height (m)			BM ^1^ (kg)			BMI ^2^ (kg/m^2^)			Spine Length (cm)			Menarche	
	Pre ^3^	Post ^4^	*p* ^5^	Pre	Post	*p*	Pre	Post	*p*	Pre	Post	*p*	No	Yes
5–11	1.420	1.440	**0.00002**	34.73	35.23	0.314	16.67	16.67	0.984	45.18	44.96	0.815	*n* = 8 (29%)	*n* = 2 (7%)
	±0.175	±0.177		±11.83	±11.66		±1.910	±1.819		±4.724	±5.002			
12–16	1.630	1.640	**0.0008**	51.22	52.16	**0.004**	19.32	19.41	0.390	51.22	51.08	0.79	*n* = 6 (21%)	*n* = 12 (43%)
	±0.064	±0.064		±8.16	±8.02		±2.574	±2.430		±3.755	±4.192			

^1^ BM—body mass; ^2^ BMI—body mass index; ^3^ Pre—before treatment; ^4^ Post—after treatment; ^5^
*p*—probability. Statistically significant differences in all Tables have been indicated in bold.

**Table 3 medicina-55-00254-t003:** Mean and SD of the angle of trunk rotation (ATR) values before (Pre) and after (Post) SKOL-AS^®^ postural training.

Variable Age (Years)	ATR [^O^]		*p*
	Pre	Post	
5–11	5.5	3.0	**0.0002**
	±2.07	±2.45	
12–16	5.2	3.0	**0.00007**
	±2.11	±1.21	

Abbreviations same as in Table 2.

**Table 4 medicina-55-00254-t004:** Mean and SD of the anthropometric points asymmetry in the posterior view before (Pre) and after (Post) SKOL-AS^®^ postural training in standing and sitting positions.

Variables Age (Years)	Posterior View Asymmetry (°)											
	Shoulder		*p*	Scapular		*p*	Pelvic		*p*	PSIS ^1^		*p*
	Pre	Post		Pre	Post		Pre	Post		Pre	Post	
	**Standing Position**											
5–11	0.720 ± 1.849	1.440 ± 2.203	0.261	0.470 ± 3.994	3.200 ± 3.373	**0.002**	0.250 ± 2.120	−0.270 ± 2.610	0.567	−0.480 ± 2.405	0.100 ± 2.130	0.431
12–16	−0.770 ± 1.839	−0.150 ± 1.787	0.183	−0.210 ± 3.933	0.960 ± 4.476	0.262	−0.400 ± 2.015	0.630 ± 1.434	0.135	−0.780 ± 2.150	−0.810 ± 1.636	0.958
*p*	0.271	0.218		0.979	0.561		0.877	0.990		0.845	0.728	
	**Sitting Position**											
5–11	1.850 ± 1.675	0.650 ± 1.699	**0.049**	1.470 ± 3.612	3.290 ± 3.521	**0.033**	1.510 ± 2.346	1.510 ± 2.645	0.998	0.500 ± 3.266	0.860 ± 2.895	0.657
12–16	0.230 ± 1.925	−0.210 ± 1.512	0.442	1.500 ± 3.568	1.010 ± 3.609	0.544	−0.080 ± 1.358	0.940 ± 1.461	**0.037**	−0.130 ± 2.391	−0.560 ± 2.342	0.056
*p*	0.104	0.647		0.999	0.454		0.225	0.901		0.570	0.208	

^1^ PSIS—posterior superior iliac spine.

**Table 5 medicina-55-00254-t005:** Mean and SD of the anthropometric points asymmetry in the anterior view before and after SKOL-AS^®^ postural training.

Variables Age (Years)	Anterior View Asymmetry (°)														
	Eye		*p*	Mouth		*p*	Clavicle		*p*	ASIS ^1^		*p*	Radial Styloid Process		*p*
	Pre	Post		Pre	Post		Pre	Post		Pre	Post		Pre	Post	
	**Standing Position**														
5–11	−1.080 ± 4.082	0.560 ± 2.873	0.077	−1.740 ± 3.872	−1.610 ± 3.997	0.909	−0.690 ± 2.2280	−0.380 ± 2.087	0.510	2.050 ± 1.802	2.060 ± 1.512	0.993	−1.190 ± 3.070	−1.110 ± 2.709	0.690
12–16	−0.980 ± 4.459	−0.140 ± 3.222	0.313	−0.500 ± 4.049	0.100 ± 3.548	0.444	0.350 ± 2.481	0.640 ± 1.843	0.436	0.110 ± 1.243	1.120 ± 1.884	**0.008**	0.070 ± 1.558	0.240 ± 1.166	0.598
*p*	0.953	0.571		0.434	0.252		0.277	0.191		**0.003**	0.184		0.171	0.111	

^1^ ASIS—anterior superior iliac spine.

**Table 6 medicina-55-00254-t006:** The mean and SD of the length of variables (left and right side of the body) and varus–valgus angle (heel–popliteus fossa) before and after SKOL-AS^®^ postural training during evaluation in the standing position.

Variables		Age (Years)	Left			Right		
			Pre	Post	*p*	Pre	Post	*p*
Length (mm)	Acromion–Heel	5–11	1199.850 ± 33.211	1212.430 ± 32.630	**0.014**	1197.100 ± 33.501	1205.720 ± 32.143	0.084
		12–16	1384.350 ± 27.537	1387.860 ± 27.056	0.682	1385.490 ± 27.778	1388.950 ± 26.652	0.625
	Acromion–PSIS ^1^	5–11	341.660 ± 40.274	340.080 ± 9.402	0.522	336.190 ± 41.388	332.500 ± 9.385	0.183
		12–16	427.780 ± 9.402	385.100 ± 7.796	0.642	423.740 ± 34.317	379.670 ± 7.781	0.609
	Acromion–Iliac crest	5–11	273.250 ± 7.302	284.420 ± 7.723	**0.033**	271.440 ± 35.07	281.330 ± 8.627	0.050
		12–16	312.290 ± 6.055	319.470 ± 6.404	**0.003**	351.730 ± 29.076	322.450 ± 7.153	0.056
	Iliac crest–Heel	5–11	926.750 ± 54.111	928.640 ± 26.905	0.999	925.950 ± 53.525	924.870 ± 61.466	0.999
		12–16	1024.050 ± 44.866	1068.570 ± 22.308	0.408	1023.740 ± 44.381	1004.290 ± 50.965	0.098
	PSIS–Heel	5–11	866.060 ± 122.502	880.650 ± 122.099	**0.00002**	867.280 ± 123.607	879.900 ± 121.992	**0.0001**
		12–16	969.720 ± 172.643	1014.810 ± 45.290	0.283	971.040 ± 168.358	1017.380 ± 44.606	0.258
	Varus–Valgus	5–11	188.210 ± 0.999	189.010 ± 1.007	0.259	185.560 ± 0.991	186.500 ± 0.928	0.240
		12–16	188.070 ± 0.828	188.690 ± 0.835	**0.018**	185.780 ± 0.822	186.970 ± 0.769	**0.0002**
	Heel–Popliteus Fossa	5–11	414.040 ± 33.872	409.410 ± 12.161	0.454	416.600 ± 33.781	408.340 ± 12.216	0.168
		12–16	511.280 ± 28.085	476.050 ± 10.084	0.438	511.240 ± 28.010	478.880 ± 10.129	0.796
Angle (°)	Heel–Popliteus Fossa	5–11	188.210 ± 16.051	189.010 ± 1.007	0.259	185.560 ± 16.395	186.500 ± 0.928	0.240
		12–16	204.850 ± 13.308	188.690 ± 0.825	0.205	203.000 ± 13.594	186.970 ± 0.769	**0.011**

^1^ PSIS—posterior superior iliac spine.

**Table 7 medicina-55-00254-t007:** The length variables before and after SKOL-AS^®^ postural training in the evaluation in sitting position.

Variables		Age (Years)	Left Side			Right Side		
			Pre	Post	*p*	Pre	Post	*p*
Length (mm)	Acromion–Iliac crest	5–11	273.120 ± 9.195	272.610 ± 8.047	0.925	271.200 ± 8.264	276.050 ± 8.147	0.408
		12–16	315.530 ± 7.624	310.330 ± 6.673	0.594	314.030 ± 6.852	314.940 ± 6.755	**0.020**
	Acromion–PSIS ^1^	5–11	357.180 ± 10.604	352.030 ± 10.731	0.180	351.380 ± 9.901	348.640 ± 10.851	0.630
		12–16	409.610 ± 8.792	406.600 ± 8.897	0.326	404.810 ± 8.209	401.550 ± 8.997	0.425

^1^ PSIS—posterior superior iliac spine.

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
