# Peer review of "The Effect of an Innovative Biofeedback SKOL-AS® Treatment on the Body Posture and Trunk Rotation in Children with Idiopathic Scoliosis—Preliminary Study"

_medicina, 2019, doi:10.3390/medicina55060254_

Round 1
Reviewer 1 Report
I suggest to better explain the study design, Preliminary study in not enough.
I suggest to specify a primary outcome as an index of therapeutic outcome.
I suggest giving some more information on the statistical results of the ATR if I am not mistaken, there is a slightly higher measure than the measurement error of the scoliometer 2°
I suggest giving other limits beyond sampling such as the absence of a control group.
Author Response
Please find attached PDF with reply to reviewer.

Reviewer 2 Report
- The format of representing numbers should be revised. For example, presenting ATR angle with 4 decimal is not correct.
- Line 28: The term "increase in height" is not clear, do you mean height of the subject?
The increase in height could be due to normal growth, because patients are in growing age. Please comment.
-Line 60: I don't agree with this statement. Having more effective stimulating would be better than having greater number of factors stimulating the body.
- Line 61: reference is required.
- Line 62: Ref 9 was published in 2001. So it is not relatively a recent work.
- Line 64: the tense of sentences should not consistent.
- Line 70 : AIS was defined before.
- Line 73: reference is needed.
In general, I noticed that references are given at the end of each paragraph, which is not a correct format of referencing the document. Please revise the whole manuscript and use references at the end of corresponding sentence.
- Line 91: citing Table 1 is enough. You don't have to mention Result section.
- Line 94: total portion of group A and B is 110%!!! Please clarify and mention the number of patients in each group.
Line 131: in Table1, height is presented with accuracy of 0.01m, which is 1cm. However, author mentioned that the height was measured with accuracy of 0.1cm. Please clarify.
- Line 132: you can explain BMI using an equation
- Line 135: Please check the heading title
- Line 145: reference is required for validation of Spinal Meter
- Line 250: What type of force and how it was applied? Did you measure the force? Who applied the force?
- Line 269: Fig6 does not have lable (a) and (b). Second image in figure 6 does not show the posture and direction of applied force clearly.
- Line 341: the caption of Table 3 indicates that reported parameters are the degree of asymmetry. I see increase of all parameters, which means that the posture became more asymmetric after training. Please clarify.
- Line 353: Why the asymmetry of eyes was measured? did you expect a difference?
- Line 384: IS was used before, no need to redefine it
-Table 5: the PSIS score for left side was:
pre 866±46
post 880±25
The SD bar contains the mean of other group. Please rerun the statistical analysis and make sure that you reported correct p value.
- Discussion: Based on the results presented in Table 2-6, the training was effective mainly on ATR, and for 5-11 age group in sitting position, and the training has no effect on other parameters. I expected a discussion about parameters with non-significant differences.
- Discussion : The discussion section does not contain strong discussions about the finding of the study. The cause of findings was not investigated.
Line 410: " there were many positive values of this research" , such as???
Line 411: The author should conduct a power analysis to make sure the sample size of 13 in 5-11, and 15 in 12-16 group is enough. If sample size is not large enough, non of statistical significant differences is reliable.
Author Response
Please find attached PDF with reply to reviewer 2.

Reviewer 3 Report
Title, line 80, 127, etc: the idea of SCOL-AS device is not novel
line 21- there no data about the speed of progression
- There is no follow-up
line 32 - abbreviation PSIS - is not explained
line 42- "Scoliosis is a spinal deformity consisting of lateral curvature in frontal plane (with a Cobb angle of 10 or more) and rotation of the vertebrae in transverse plane [1,2]" - the scoliosis is 3D deformation- in the sagittal plane as well.
line 3, 20, 73, etc. SKOL-AS is an abbreviation? Company? Method? Device?
line 75- 76- "The patient immediately receives a feedback signal about the correctness of the exercises performed, which significantly speeds up the learning process of correct patterns." - this is a conclusion without proof- it can be authors expectation only.
line 85- the group is very heterogenic. Not acceptable.
How the children were qualified for this kind of treatment?
Lack of table with all patients data: age, gender, Risser, levels, Cobb angle, etc. to make the comparing data easier for the reader.
line 92- "Because there were statistically significant differences between younger and older participants...," - what kind of differences? p-value?
Line 93 - "most of the children " - "most" is not a scientific word - how many?
line 96- " Risser stage (RS) showing higher progression incidence" RS shows skeletal maturity, not a progression incidence
line 97- Risser 0 group should be separately evaluated - there is no sign of skeletal maturation at all, and children are before the growth spurt, In Risser 2 - after
line
line 97- "thus" seem to be not a proper word, because of in all children ", the proper diagnosis and immediate physiotherapy should be implemented"
line 107- too heterogenic group- only one type should be analyzed, or patients from each group should be compared. No control- eg patients with “posture correction” only
line 122 all data should be given in one table
line 146 add a reference to that statement
Line 215- the hight of 6 and 16 y.o.differ. What was the hight of the camera stand?
Line 233 – in 2 curves patients – which ATR was taken into account? If both, there is no data in results
Line 237 – putting a scoliometer on the “same level as in preliminary examination “seems to be questionable while a patient is dressed
Line 242- when control examination was performed? Immediate after a session? Maybe the result is temporary?
Line 246 – the follow up in next 3 months would be very valuable – permanents of effect- (line 242) then an x-ray could be taken as well.
Line 250- treatment of scoliosis should be individualized. It seems that all patients had the same protocol, doesn’t matter eg. the age, or kind of curvature
Line 252-How the level of the apex of the curve was calculated (which vertebrae)Was it marked on the skin? – It can’t be seen in a dressed child. How big force was applied?
Line 260-261 not clear: “The effective exercise work performed by the patient included the deep muscles exercise, visual biofeedback starting from 40 mmHg to e.g. 60, 70, 80, 100 mmHg on manometer scale.” When/what protocol to change the pressure?
Line 275 What was the protocol for patients in “no SKOL-AS” days?
Line 301- what p-value is statistically significant?
Line 305 – age 5- in material – age 6
Line 325 Table 1- age 5- in material – age 6
Line 325: Spine length – haw it was measured? X-ray? Which levels? Sum of each vertebra? T1-S1 spinal process distance?
Line 390- How the Author explain increase the height with no spinal length difference? Maybe it is a head position? Especially that spine length was shorter post vs pre (with no statistical significance). It can suggest progression of the deformation. Maybe the higher the height is because of the growth of the child.
Line 391- that is why sentence: “This confirmed positive effect of elongation exercise used in SKOL-AS® device treatment.“ is too optimistic.
Line 405- there is no data to prove it – no x-ray.
Line 406 – only a few parameters (ATR, shoulder in sitting improved. “Scapular” Pelvic in sitting and ASIS asymmetry increased.
Line 419- not comes from results- it does not elongate the spine.- but shorten (no Statistical significance)
Line 420- but “Scapular” Pelvic in sitting and ASIS asymmetry increased. It means that effectiveness is questionable.
Line 422- Doesn’t come from results
Line 424- as line 420 – “learning good postural habits…” is questionable
Author Response
Please find attached PDF with response to Reviewer 3 comments.

Round 2
Reviewer 2 Report
Thanks for addressing reviewers' comments.
There are still some miss-spellings that need to be addressed. For instance in Line 287.
Author Response
As suggested by the reviewer, all manuscript has been checked and revised by authors as well as the professional English teatcher. All corrections have been made (and highlighted by red color) in the attached file of the manuscript.

Reviewer 3 Report
Line 4 – 3 months f-up is still a preliminary study
Line 531 The permanence of the effect is not clear, so it is better to write “seems to be” against “is a good way”
Line 420 of my first review - asymmetry of this parameter increased - so the effectiveness is questionable not a promising as authors suggest
Author Response
1. Line 4 – 3 months f-up is still a preliminary study
Response:
We appreciate the comment about the follow-up study. The tile has been changed for:
The effect of an innovative biofeedback SKOL-AS® treatment on the body posture and trunk rotation in children with idiopathic scoliosis - preliminary study.
2. Line 531 The permanence of the effect is not clear, so it is better to write “seems to be” against “is a good way” .
Line 420 of my first review - asymmetry of this parameter increased - so the effectiveness is questionable not a promising as authors suggest
Response:
As suggested by the reviewer, the conclusions has been changed. The suggested corrections now reads:
· The SKOL-AS® biofeedback method uses the three-dimension exercise, thus it seems to be a good way to treat scoliotic spine.
· The decrease in trunk rotation and some anthropometric points asymmetry and the increase in length values proves the effectiveness of the SCOL-AS® treatment. However, some results are questionable and need verification.
